

# 1 Method to calculate the aerosol asymmetry factor based on measurements from

# 2 the humidified nephelometer system

**Gang Zhao[1], Chunsheng Zhao[1], Ye Kuang[2], Yuxuan Bian[3], Jiangchuan Tao[2], Chuanyang Shen[1],**
**Yingli Yu[1]**
[1]Department of Atmospheric and Oceanic Sciences, School of Physics, Peking University, Beijing,
China
[2]Institute for Environmental and Climate Research, Jinan University, Guangzhou 511443, China
[3]State Key Laboratory of Severe Weather, Chinese Academy of Meteorological Sciences, Beijing,
100081, China
Corresponding author: Chunsheng Zhao (zcs@pku.edu.cn)
**Abstract**
The aerosol asymmetry factor (g) is one of the most important factors for assessing direct aerosol
radiative forcing. So far, few studies have focused on the measurements and parameterization of g. The
characteristics of g are studied based on field measurements over the North China Plain by using the
Mie scattering theory. The results show that calculated g values can vary over a wide range (between
0.54 and 0.67). When ambient relative humidity (RH) reaches 90%, g is significantly enhanced by a
factor of 1.2 due to aerosol hygroscopic growth. Direct aerosol radiative forcing can be reduced by 40%
when g increases by 20%. For the first time, a novel method to calculate g based on measurements
from the humidified nephelometer system is proposed. This method can constrain the uncertainty of g
within 2% for dry aerosol populations and 4% for ambient aerosols, taking into account aerosol
hygroscopic growth. Sensitivity studies show that ambient RH and aerosol hygroscopicity are the most
important factors that influence the accuracy of predicting g.
**1 Introduction**
In addition to aerosol optical depth and aerosol single-scattering albedo, the aerosol phase function
is the most important factor for assessing direct aerosol radiative forcing (DARF). The
Henyey-Greenstein (HG) phase function is a widely used method to parameterize the phase function
because it uses the aerosol asymmetry factor (g) as the only free parameter. The HG phase function is
expressed as

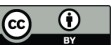



$$P_{HG}(\theta) = \frac{1-g^2}{(1+g^2-2g\cos\theta)^{3/2}} \tag{1}$$

where $\theta$ is the angle between the incident light direction and the scattered light direction. In this
respect, the free parameter g can reflect the angular aerosol scattering energy distribution.
g is defined as:
$$g = \frac{1}{2}\int_0^\pi \cos\theta P(\theta)\sin(\theta)\,d\theta \tag{2}$$

where P(θ) is the aerosol phase function. As a result, g can be a computationally efficient factor
that replaces the phase function to study aerosol radiative transfer properties (Toublanc, 1996;Hansen,
1969;Boucher, 1998). Some researchers have widely accepted the use of g as a replacement of the
phase function (Hansen, 1969). However, the g-related HG phase function may cause significant bias
when estimating photodissociation rates (Toublanc, 1996) and aerosol radiative forcing effects
(Boucher, 1998). Up to now, there have been few studies that have assessed the bias when replacing
the ambient phase function with the g-related HG phase function by using field measurements of
aerosol optical properties. Moreover, variations in g can influence the evolution of the atmospheric
vertical structure through its effects on the atmospheric radiative distribution. Kudo et al. (2016) also
found that the vertical profile of the asymmetry factor plays an important role in altering vertical
variations in the solar heating rate. Marshall et al. (1995) reported that a 10% overestimation of g can
systematically reduce aerosol climatic forcing by 12% or more. Andrews et al. (2006) found that a 10%
reduction in g would result in a 19% overestimation of atmosphere radiative forcing at the top of
atmosphere (TOA). An accurate estimation of g can help improve the assessment of the aerosol
radiative effect.
Though many methods can be used to derive g, there is no available method to measure g directly.
Horvath et al. (2016) measured the phase function of aerosols, calculated the g of aerosols, and found
that the HG phase function can be used as a good approximation of the measured phase function.
Many works used the Mie model (Bohren and Huffman, 2007) to calculate the phase function and
proved its reliability (Andrews et al., 2006;Marshall et al., 1995;Shettle and Fenn, 1979;Bian et al.,
2017). Comprehensive attempts have been made to relate g with the hemispheric backscatter fraction
(b), where b is the ratio of light scattered into the backward hemisphere compared to total light
scattered in all directions (Wiscombe and Grams, 1976;Andrews et al., 2006).





The free parameter g varies significantly for different aerosol types and different seasons.
D'Almeida et al. (1991) suggested that g at a wavelength of 500 nm ranges from 0.64 to 0.83
depending on the aerosol type and season. A mean value of 0.67 at an ambient RH was also
recommended (D'Almeida et al., 1991). Hartley and Hobbs (2001) reported a median g value of 0.7 for
aerosols along the east coast of the United States. Formenti et al. (2000) measured Saharan dust
aerosol and found that the aerosol g values ranged from 0.72-0.73. Biomass burning aerosols in Brazil
had a low g value of 0.54 (Ross et al., 1998).
Some works have studied the impacts of aerosol hygroscopic growth on the parameter g (Hartley
and Hobbs, 2001;Kuang et al., 2015) and found that variations in g with RH can have significant
influences on aerosol radiative effects (Kuang et al., 2015;Kuang et al., 2016). A parameterization
scheme of g, that takes RH and aerosol hygroscopic growth into account, is necessary.
When exposed to the ambient atmosphere, aerosols can grow by absorbing water, which causes
their corresponding optical properties to considerable change. The κ-Köhler theory (Petters and
Kreidenweis, 2007) is widely used to describe the hygroscopic growth of aerosol particles by using a
single aerosol hygroscopic growth parameter (κ) and the κ-Köhler equation, which is shown as
$$\frac{RH}{100} = \frac{gf^3 - 1}{gf^3 - (1-\kappa)} \cdot \exp(\frac{4\sigma_{s/a} M_{water}}{R \cdot T \cdot D_d \cdot gf \cdot \rho_w}) \qquad (3)$$
where $D_d$ is the dry particle diameter; gf(RH) is the aerosol growth factor, which is defined as the
ratio of the aerosol diameter at a given RH and the dry aerosol diameter ($D_{RH}/D_d$); T is the
temperature; $\sigma_{s/a}$ is the surface tension of the solution; R is the universal gas constant and $\rho_w$ is the
density of water. The aerosol hygroscopic growth parameter κ can be further used to investigate the
influence of aerosol hygroscopic growth on aerosol optical properties (Tao et al., 2014;Kuang et al.,
2015;Zhao et al., 2017) and aerosol liquids water contents (Bian et al., 2014).
According to the Mie theory, g is associated with aerosol particle number size distribution, the
particle complex refractive index, the aerosol mixing state and ambient RH. Datasets from the
humidified nephelometer system can partially account for all of these factors. The humidified
nephelometer system consists of two nephelometers: one nephelometer measures dry aerosol scattering
properties and the other measures aerosol scattering properties under well-controlled RH conditions.
This results in the light scattering enhancement factor (fRH), which is defined as fRH(λ,RH)=
$\sigma_{sca(\lambda,RH)}/\sigma_{sca(\lambda,dry)}$, or the ratio of the aerosol scattering coefficient under given RH conditions to that of



dry conditions. Each nephelometer can provide a scattering coefficient ($\sigma_{sca}$) and back-scattering
coefficient ($\beta_{sca}$) at three wavelengths (450, 525, 635nm). $\sigma_{sca}$ can be used to calculate the aerosol
scattering Ångstrom index, which reflects the aerosol particle numbers size distribution (PNSD) to
some extent. In general, a larger value for the Ångstrom index always corresponds to a smaller
predominant aerosol size. Variations in $\beta_{sca}$ and $\sigma_{sca}$ can be used to deduce the aerosol BC mixing state
(Ma et al., 2012). At the same time, datasets from the humidified nephelometer system can also be
used alone to measure the aerosol hygroscopicity and provide an overall hygroscopic parameter κ
(Kuang et al., 2017). All in all, measurements from the humidified nephelometer system might be used
for estimating g under the given RH conditions.

In this study, the Mie scattering theory and field measurements over the North China Plain (NCP)

are used to study the characteristics of g. Section 2 describes the related datasets used in this study.
Details of the study on the characteristics of g and impacts of aerosol hygroscopic growth on g are
shown in section 3.1. A new method, which is based on a random forest machine learning model, is
introduced to calculate g in section 3.2. We also discuss the impacts of g variations on the uncertainties
of DARF in section 3.3, and the corresponding results are presented in section 4.3. Section 4.1 gives
the calculated characteristics of g and section 4.2 proves the feasibility of using the machine learning
model to calculate g. Finally, this method is validated by the ambient aerosol phase function measured
with a charge-coupled device -laser aerosol detective system (CCD-LADS) in section 4.4. Conclusions
are in section 5.
**2. Instruments and datasets**

Datasets used in this study come from three field campaigns, which were conducted at three

different sites in the NCP. These three campaigns are conducted at the AERONET BEIJING_PKU
station in Beijing (PKU), Gucheng in Hebei Province (Gucheng), and the Yanqi Campus of the
University Of Chinese Academy of Sciences in the Huairou district, Beijing (Huairou) and these
locations are shown in Fig. S1. The PKU station is located at the northwest of Beijing, between the 4[th]
and 5[th] ring road. Datasets for this location are representative of urban aerosols in the NCP. Gucheng is
located between two megacities (120 km from Beijing and 190 km from Shijiazhuang) of NCP and the
pollution conditions of Gucheng can be a good representation of the continental background in the
NCP. Details for the Gucheng station can be found at Kuang et al. (2017). The UCAS station is 60 km



away from the center of Beijing and is at the edge of the NCP, which makes it suitable for measuring
the regional pollution properties of the NCP (Ma et al., 2016).

Table 1 lists the information for the field campaigns and the datasets used in this study. During the

campaigns, sampled aerosols that had an aerodynamic diameter of less than 10 µm by an impactor at
the inlet are selected. These aerosols are then dried to below 30% RH with a Nafion drying tube and
then lead to each instrument. Aerosol PNSDs ranging from 3 nm to 10 µm are measured by using the
scanning mobility particle size spectrometer (SMPS, TSI Inc., model 3936) and an aerodynamic
particle sizer (APS, TSI Inc., model 3321) with a temporal resolution of 5 min. Black carbon (BC)
mass concentrations are measured by a multi-angle absorption photometer (MAAP model 5012,
Thermo, Inc., Waltham, MA USA) at Huairou and by an Aethalometer 33 (Hansen et al.,
1984;Drinovec et al., 2015) at PKU and Gucheng. The aerosol $\sigma_{sca}$ at wavelengths of 450 nm, 525 nm
and 635 nm is measured by an Aurora 3000 nephelometer (Müller et al., 2011). The f(RH) is measured
by a self-constructed humidified nephelometer system with a time resolution of approximately 45 min.

Ambient aerosol phase function is measured at Huairou by using a CCD-LADS. This system

consists of a continuous laser, two charge-coupled device cameras and the corresponding fish eye
lenses. It can measure the ambient aerosol phase function at a wide angular range of 10-170$^{\circ}$ with a
high resolution of 0.1$^{\circ}$. More details of the measurement system can be found at (Bian et al., 2017).
**3. Methodology**
**3.1 Calculating characteristics of g based on the Mie scattering theory**

The Mie model (Bohren and Huffman, 2007) is employed to calculate the characteristics of g. Its

results include aerosol phase function, and g can be calculated by the definition shown in formula 2.
When running the Mie model, aerosol PNSD and BC are necessary. The complex refractive index of
non-absorbing aerosols is $1.53+10^{-7}$i (Wex et al., 2002a) at a wavelength of 525 nm. BC is treated as
partially internal mixed and the remaining aerosols are treated as core-shell mixed. The ratio of
internally mixed BC mass concentrations to core-shell mixed BC mass concentration is determined by
the method introduced in Ma et al. (2012). The size distribution of BC mass concentration, which is
adopted from Ma et al. (2012) is also used in this work. The density and refractive index of BC are set
as 1.5 g/cm$^3$ and 1.8+0.54i (Kuang et al., 2015), respectively. With this information, the value of g
under each measured PNSD at dry state can be calculated.

The κ-Köhler theory and the Mie scattering model are employed to calculate g under different RH



conditions. The real time κ, which is derived from the measurement of f(RH), is used to account for
aerosol hygroscopic growth. For each RH value, the gf can be calculated based on formula 3. The
corresponding ambient aerosol PNSD at a given RH can be determined too. The refractive index ($\tilde{m}$),
which accounts for water content in the particle, is derived as a volume mixture between the dry
aerosol and water (Wex et al., 2002b):

$$\tilde{m} = f_{V,dry}\, \tilde{m}_{aero,dry} + (1 - f_{V,dry})\, \tilde{m}_{water} \qquad (4)$$

where $f_{v,dry}$ is the ratio of the dry aerosol volume to the total aerosol volume under a given RH
condition; $\tilde{m}_{aero,dry}$ is the refractive index for dry ambient aerosols and $\tilde{m}_{water}$, the refractive index
of water, is $1.33+10^{-7}i$. Then, the corresponding g values under the given RH and PNSD can also be
calculated. More details on using the Mie model to calculate the aerosol phase function for different
RH conditions can be found in Zhao et al. (2017).

**3.2 Calculating g by using the random forest machine learning model**

The random forest model consists of a set of decision tress, and then, a simple majority vote from
the decision tree is used for prediction. This model is a widely used nonparametric machine learning
algorithm that has several strengths. First, it involves fewer assumptions regarding the dependence
between observations and outcomes when compared with traditional parametric regression models.
Second, strict relationships among variables are not needed before implementing the random forest
model.
The random forest model has two parameters: the number of input variables ($n_{pre}$) and the number
of trees grown ($n_{tree}$). In this study, $n_{pre}$ and $n_{tree}$ are set as eight and ten, respectively. The eight input
parameters include the three scattering coefficients, three backscattering coefficients, RH and κ. For
these parameters, the ambient RH value come from automatic weather station, and the rest of the data
come from the humidified nephelometer system measurements. The predictor g values are calculated
from Mie scattering results.
The measured datasets are divided into two parts: one for the training data of the random forest
model, and the other for test data. All training datasets come from field measurements at AERONET
BEIJING_PKU station, whereas the datasets from Gucheng are employed to test the accuracy of the
model. With split datasets from different sites, the feasibility of the random forest model in the NCP
can be guaranteed. Before calculating g, we compare the measured $\sigma_{sca}$ from the nephelometer and
calculate $\sigma_{sca}$ from the Mie scattering model. These data, where the relative difference between the



measured and calculated $\sigma_{sca}$ is within 30%, are used for the following analyses. More details regarding
the used data are shown in section 1.2 of the supplementary material.

**3.3 Aerosol DARF estimations**

The earth-atmosphere systems can be significantly influenced by aerosols, which scatter and
absorb the energy. In this study, the Santa Barbara DISORT (discrete ordinates radiative transfer)
Atmospheric Radiative Transfer (SBDART) model (Ricchiazzi et al., 1998) is employed to estimate
the DARF. The characteristics of DARF with the variations in g are studied.
The instantaneous DARF is calculated at the TOA for cloud-free conditions. DARF is defined as
the difference between radiative flux at the TOA under present aerosol conditions and aerosol-free
conditions:

$$\text{DARF} = (f_a \downarrow - f_a \uparrow) - (f_m \downarrow - f_m \uparrow) \qquad (5)$$

where $(f_a \downarrow - f_a \uparrow)$ is the downward radiative irradiance flux with given aerosol distributions and
$(f_m \downarrow - f_m \uparrow)$ is the radiative irradiance flux under aerosol free conditions. The wavelengths for
irradiance range from 0.25 to 4 µm.
Input data for the SBDART are listed below. Vertical profiles of the aerosol optical properties,
which include the aerosol extinction coefficient ($\sigma_{ext}$), aerosol single scattering albedo (SSA) and g
with a height resolution of 50 m, come from the parameterization of aerosol vertical distributions and
the results of the Mie model. Methods for parameterization and calculation of the aerosol optical
profiles can be found in Kuang et al. (2016) and Zhao et al. (2017). Atmospheric gas and
meteorological parameter profiles come from the mean results of the radiosonde observations at the
Meteorological Bureau of Beijing (39°48' N, 116°28' E), which include profiles for water vapor,
pressure and temperature during the summer. Surface albedo values are obtained from the Moderate
Resolution Imaging Spectroradiometer (MODIS) V005 Climate Modeling Grid (CMG) Albedo
Product (MCD43C3) during December, 2016 of Gucheng, where one of the field campaigns is
conducted. The remaining input data for the SBDART are set to their default values.

**4 Results and Discussion**

**4.1 Characteristics of g**

**4.1.1 Characteristics of g at different sites**

Fig. 1 gives the statistical results for the calculated g properties at Gucheng and PKU. The RH at
the two sites shows almost the same diurnal variation pattern in Fig. 1 (a) and (b). The RH reaches a



peak in the morning at approximately 6:00 am, and then reaches its lowest value at approximately
16:00 in the afternoon. However, the mean values of RH are 77.7%±20.9% at Gucheng and
47.8%±20.8% at PKU. Ambient air is obviously drier during the spring than in winter in the NCP. The
g values under dry conditions that are calculated by a measured PNSD have almost no diurnal patterns.
The g values at Gucheng (0.601±0.021) are slightly lower than those at PKU (0.614±0.025) as shown
in Fig. 1 (c) and (d). The difference in g values results from different aerosol properties at these two
sites. However, ambient g values have different patterns at different sites, as shown in Fig. 1 (e) and (f).
The g values have an RH-related diurnal pattern at Gucheng, with a mean value of 0.668±0.073, but
show no diurnal variation at PKU, where the mean value of g is 0.615±0.028. The g value is
significantly influenced by RH when the RH is higher than 80%, which will be detailed in section
4.1.2. Ambient g values at Gucheng can vary from 0.57 to 0.8, comparable to those of Andrews et al.
(2006), which range from 0.59 to 0.72.
**4.1.2 Influence of RH on g**
To assess the influence of RH on g, the g values are calculated under different RH conditions for
each aerosol PNSD. The statistical results of g versus RH are shown in Fig. 2. The g value has a mean
of 0.61 at dry conditions and can vary widely (0.54 to 0.67), which corresponds to approximately 10%
of the variation. However, the mean g value can vary from 0.65 to 0.8 when the RH reaches 90%. The
g enhancement factor, which is defined as the ratio of g at a given RH and g under dry conditions, can
reach a mean value of 1.2 at an RH of 90%, which means that the g value under wet conditions is
approximately 20% higher than that under the dry conditions. This finding is consistent with that of
Hartley and Hobbs (2001), who found that g is highly related to the RH.
**4.2 Calculating g by using the machine learning model**
**4.2.1 Feasibility of using the random forest model**
We establish two independent random forest machine learning models to predict g values under
dry conditions and under ambient RH conditions separately.
When running the random forest machine learning model, $\sigma_{sca}$ and $\beta_{sca}$ at three different
wavelengths, measured by an Aurora 3000 nephelometer, are used as the input for independent
variables. The other two input parameters, RH and $\kappa$, are set equal to zero when predicting g values
under dry conditions. The predictor g values come from the results of the Mie scattering model. Fig.
3(a) shows the calculated g values and the random forest model predicted g values under dry



conditions. The results show that the calculated and predicted g values show good consistency with an
$R^2$ value of 0.90. The corresponding deviations in the predicted g values from calculated g values are
only 2.3%.
To predict ambient g values, the RH is set at ambient values, and κ is set to concurrently derived
values from the humidified nephelometer system. Fig. 3(b) shows the results of the predicted ambient
g values and g values calculated by the Mie scattering model. The correlation coefficient reaches 0.94
with a standard deviation of 4.2%. The random forest model can be a good method to predict g values.
The filled colors of the dots in Fig. 3 represent the concurrently measured $\sigma_{sca}$. It is shown that
with an increase in $\sigma_{sca}$, g values tend to be larger. When a particle has larger diameters, the $\sigma_{sca}$ of the
particle is higher, and there tends to be a larger partition of forward scattering light.
Wiscombe and Grams (1976) studied the relationship between b and g and gave the expression
between them as follows:
$$g = -7.143889 \cdot b^3 + 7.464439 \cdot b^2 - 3.96356 \cdot b + 0.9893 \quad (6).$$
This equation is widely used to calculate g from b (Andrews et al., 2006;Horvath et al.,
2016;Kassianov et al., 2007). We use the field measurement results to test its reliability. The
comparison results between calculated g values from the Mie scattering model and parameterized g
values from equation 5 are shown in Fig.S3. From Fig.S3, we can see that the parameterized g values
are prevalently larger than the calculated g values by approximately 10%. When the $\sigma_{sca}$ is smaller, the
deviations become larger. This result means that the previously established parameterization scheme is
not applicable in the NCP.
**4.2.2 Sensitivity of the random forest model**
Sensitivity studies are carried out to assess the influence of each input variable on g. First, we run
the random forest model with measured input variables and record predicted g values. These g values
are marked as $g_0$. Second, input test variables are randomly increased or decreased by 5% percent of
the measured values and are used as new input variables. Then, the input variables are used to predict g
values, which are compared with $g_0$. Finally, each variable is changed by 10% and 20% separately, and
deviations of the predicted g are studied.
Fig. 4 gives the results of the deviations of g due to variations in each variable. The results show
that when the input test variables change by 5%, the predicted g values are mainly sensitive to the
given κ with a variation of 1.74%. A 5% variation in RH can lead to a 1.17% variation in predicted g.



However, g is not sensitive to the single measured $\sigma_{sca}$ and $\beta_{sca}$ variables. This finding might be
explained by the high correlation between the measured $\sigma_{sca}$ and $\beta_{scs}$. Thus, the high uncertainties of
the measured $\beta_{sca}$ have little influence on the prediction of g. Variations in g corresponding to κ
increase slowly from 1.8% to 1.9% when κ changes by 10% and 20%, respectively. However,
variations in RH can reach 1.8% and 3.3%. It is concluded that κ and RH are the two most important
parameters when predicting g using the random forest model.
**4.3 Estimating the impacts of g on DARF**
**4.3.1 Uncertainties of replacing the real phase function with the HG phase function**
When the HG phase function is used to parameterize the actual phase function, there are some
deviations and the influence of these deviations should be estimated. The relative difference between
the DARF from the actual phase function and the DARF from the HG phase function is used to
estimate uncertainties when using the HG phase function. First, the actual calculated aerosol phase
function profiles are used as inputs to estimate DARFs. The phase function is then replaced with the
g-related HG phase function, and the DARFs are calculated again. These relative differences between
the DARFs from the above two steps are recorded and compared. The relative differences at different
zenith angle conditions are calculated to comprehensively estimate the influence of the HG phase
function.
Fig.5 shows the estimated DARFs at different zenith angles. In Fig. 5(a), DARF at the TOA can
vary from -2.55 to -4.8 w/m$^2$. When the phase function is replaced by the HG phase function, the
calculated DARF ranges from -2.6 to -5.1 w/m$^2$. The relative difference of the DARFs between the two
methods ranges from 1.3% to 7.1%, as shown in Fig. 5(b). It is concluded that using the g related HG
phase function to replace the actual phase function to estimate aerosol radiative effects is applicable,
with a deviation of less than 7%, in the NCP.
**4.3.2 Impacts of g variations on DARF estimation**
Variations in g can lead to significant variations in the estimated DARF (Kuang et al.,
2016;Andrews et al., 2006). We study the influence of g on DARF by increasing or decreasing by
2.3%, 10%, and 20% to the original g value, then comparing the corresponding DARFs with the
original DARFs. Fig. 6 shows the estimated DARFs with filled colors as well as relative differences in
the calculated DARF. The results show that when g varies by 2.3%, the DARF can vary at a variation
ratio of 6%. However, variations of 10% and 20% for g values can lead to relative difference in the



estimated DARF of 20% and 39%, respectively.

As shown in sections 4.1.1 and 4.1.2, g in a dry state can have a variation of 10% and when

considering aerosol hygroscopic growth, the variation in g can reach 20%. It is therefore very
important to take the variations in g into consideration when estimating the aerosol DARF.
**4.4 Validation of the random forest machine learning model**

Datasets of the Huairou campaign are used to validate the random forest machine learning model.

On one hand, the g values are calculated by using the random forest machine learning model with the
datasets of the humidified nephelometer ($g_{Machine,\ cal}$). On the other hand, ambient g values are
calculated by using the measured phase function from the CCD-LADS ($g_{CCD,cal}$) according to the
definition shown in formula 2. Then the g values calculated with these methods are compared.

Comparison results of these two kinds of g values are shown in Fig. 7. Form Fig.7, the values of

$g_{Machine,cal}$ and $g_{CCD,cal}$ show good consistence. There are 90% of the conditions that the relative
differences between the $g_{Machine,cal}$ and $g_{CCD,cal}$ are in the range of 6.6% which is a little higher than the
relative difference of the g values (4.2%) between machine learning method and the Mie scattering
method. During the period, the $\sigma_{sca}$ range from 30 to 260 $Mm^{-1}$ which lead to cleaner conditions in
Huairou than in Gucheng and PKU. Correspondingly, most of the g values are small and locate at the
range of 0.54 to 0.62 which are obviously lower than those in other campaigns. At the same time, the
surrounding condition at Huairou during the winter is relative dry, which results to small g values.
These conditions may partially explain the relatively higher difference between the $g_{Machine,cal}$ and
$g_{CCD,cal}$. With this validation, we conclude that the random forest machine learning model can give a
reasonable g value based on the measurements of the humidified nephelometer system.
**5 Conclusions**

The characteristics of g in the NCP are studied based on the Mie scattering theory and field

measurements from sites of Gucheng and PKU. The results show that g values are 0.604±0.025 at
Gucheng and 0.615±0.021 at PKU. The ambient g values at Gucheng show obvious diurnal variations
due to variations in RH. When the ambient RH reaches 90%, g can be enhanced by 20%. Comparison
of the calculated g values from the Mie scattering model and the parameterized g values from the
Wiscombe and Grams (1976) method shows that the parameterized g is overestimated by
approximately 10% and that the deviations are even greater when the measured $\sigma_{sca}$ is below 200
$Mm^{-1}$.



The random forest machine learning model and datasets from the humidified nephelometer are employed to calculate g values. The input data of the random forest model contain measured $\sigma_{sca}$ and $\beta_{sca}$ at three wavelengths, RH and the hygroscopic parameter $\kappa$. Except for RH, all input data came from measurements from the humidified nephelometer system (Kuang et al., 2017). The random forest model is able to improve the accuracy of predicting g, and the uncertainties of the predicted g values are constrained to be within 2.3% under dry conditions and 4.2% under ambient conditions. This is the first time that datasets from the humidified nephelometer system and machine learning are combined to study g. A sensitivity study of the random forest model shows that the predicted g is sensitive to ambient RH and the aerosol hygroscopic parameter $\kappa$.

The new method for calculating g is validated by comparing the g values calculated by using the random forest machine learning model and the g values calculated from the phase function measured by using the CCD-LADS. The g values with this two methods show good consistence with 90% of the data within the relative difference of 6.6%.

SBDART is used to study the impacts of g on DARF. We first studied the relative differences between estimated DARFs by using the HG phase function and the actual phase function. The results show that the relative differences in DARF can be contained within 7.1% when replacing the actual calculated phase function with g related HG phase function. The HG phase function can be a feasible parameterization scheme to study DARF in the NCP. The sensitivity study shows that variations of 10% and 20% from g can lead to variations in DARF of 20% and 39%, respectively. This work can further our understanding of the role of g in the radiative effects of aerosols and can help reduce uncertainties in estimating DARF.

**Acknowledgements**

This work is supported by the National Natural Science Foundation of China (41590872) and the National Key R&D Program of China (2016YFC020000:Task 5).

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




**Table 1.** Field information, dataset information and instruments that are used in this study.

| Field information | | Datasets and instruments | | | | |
|---|---|---|---|---|---|---|
| Location | Time period | PSND | BC | $\sigma_{sc}$ | f(RH) | Phase fucnition |
| Gucheng, Hebei (39°09' N, 115°44' E) | 15 Oct to 25 Nov, 2016 | SMP, APS | AE33 | Aurora 3000 | Humidified Nephelometer | None |
| PKU, Beijing (39°59' N, 116°18' E) | 21 Mar to 10 Apr, 2017 | SMPS, APS | AE33 | Aurora 3000 | Humidified Nephelometer | None |
| Huairou, Beijing (40°24' N, 116°40' E) | 3 Jan to 27 Jan, 2016 | SMPS, APS | MAAP | Aurora 3000 | Humidified Nephelometer | CCD-LADS |







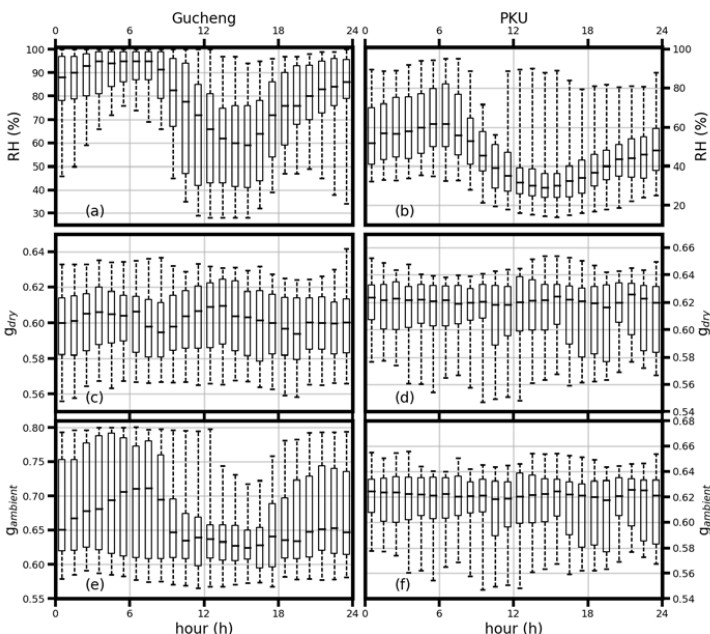


**Figure 1. (a)(b)** Average diurnal pattern of RH, **(c)(d)** g values calculated from dry aerosols, and **(e)(f)**

g values from ambient aerosols. The panels **(a), (c)** and **(e)** are the results from Gucheng. Panels **(b), (d)**

and **(f)** are the results from PKU. The box and whisker plots represent the 5[th], 25[th], 75[th] and 95[th]

percentiles.



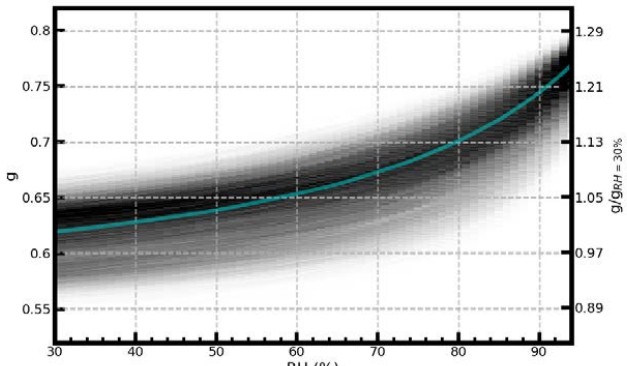

**Figure 2.** Probability distributions of g under different RH conditions. The ticks on the left show g

values at different RH values, and the ticks on the right show the g enhancement factor, which is

defined as the ratio of g at a given RH to the g value at dry conditions (RH=30%). The solid line (cyan)

shows the mean result of g values and the enhancement factor at different RH values.



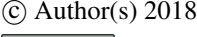




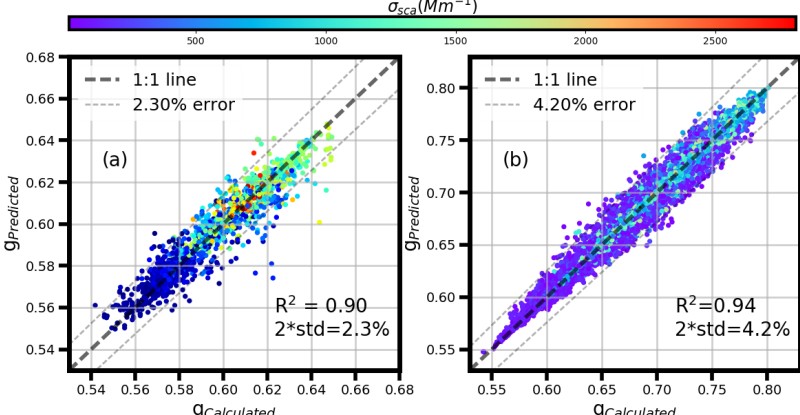

**Figure 3.** Comparison of calculated g values from the Mie model and predicted g values from the random forest model under (a) dry conditions and (b) ambient conditions. Colored dots represent the concurrently measured $\sigma_{sca}$ corresponding to the time of g.






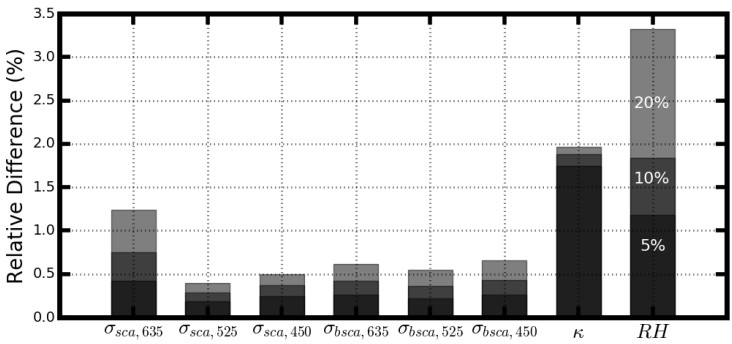

**Figure 4.** Variations in g when the input parameters vary separately by 5% (dark color), 10% (grey color), and 20% (light grey color).





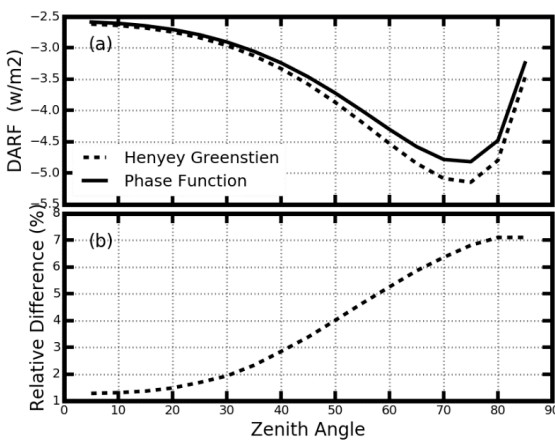

**Figure 5. (a)** Estimated DARFs at different zenith angles when using the g related HG phase function
(dotted line) and the calculated phase function (solid line). (b) The relative difference between the
DARFs in (a).




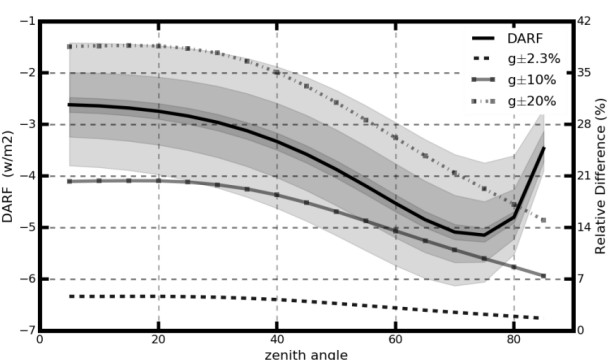

**Figure 6.** The variation in DARF when g varies by a range of 2.3% (the filled dark color), 10% (grey color), and 20% (light grey color). Different line styles represent the corresponding mean relative differences in DARF compared to the original value.





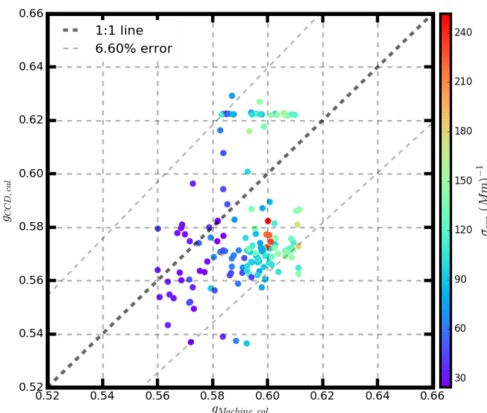

**Figure 7.** Comparison of the calculated g values ($g_{Machine,cal}$) by using the random forest machine

learning model and the calculated g values ($g_{CCD,cal}$) from the CCD-LADS measured phase fuction.