# Peer review of "Method to calculate the aerosol asymmetry factor based on measurements from"

_Atmospheric Chemistry and Physics, 2017_

## Referee Comment (RC1) · Anonymous Referee #2 · 4 Apr 2018

**Referee's Comments**
**Manuscript: acp-2017-1148**

**Summary:**
In this paper, the authors use measurements of dry and humidified aerosol properties to validate a new machine learning algorithm for determining the asymmetry parameter based on routinely measured properties. The asymmetry parameter ($g$) is important in some radiative forcing models that are based on the Henyey-Greenstein approximation of aerosol scattering phase function, a function only of $g$. The algorithm for determining $g$ using dry and humidified nephelometer measurements is validated with the CCD-LADS, a new instrument that directly measures the aerosol scattering phase function. An attempt is then made to connect these results with radiative forcing models.

**General comments:**

- It would be useful to have more information about the different aerosol properties at the three different ground sites. For example, how do the PNSD and BC fraction vary between the different sites, as well as seasonally and diurnally? In particular, examples of the PNSD (dry and ambient) averaged for the different sites would be a useful figure to have since particle size is such an important factor in determining asymmetry

- The weakest part of the study is the machine learning algorithm section. The use of the random forest algorithm is not well-justified, nor is there sufficient discussion of the parameterizations used.

- I find the label "calculated phase function" and "measured phase function", used extensively in the discussion and throughout the figures, confusing. Please clarify whether referring to theoretical phase functions determined using the PNSD, Mie theory, and assumed refractive index, or phase functions directly measured by CCD-LADS and be consistent with how these are referred to.

- I think it would strengthen the discussion to include estimations of the uncertainties of the various measured properties. These can then be used for sensitivity analysis instead of arbitrary deviations.

- The authors should be careful about the input parameters of the random forest algorithm, particularly for their sensitivity analysis. The measured quantities $\sigma_{sca}$ and $\beta_{sca}$ for a specific wavelength are related to each other, and therefore I think it is important to question whether they can be treated as independent input variables.

- Please ensure that axes labels on figures legible (text size difficult to read on e.g. Figure 7)

**Specific comments:**

P1 L15-16: Does this range only apply to dry aerosol?

P1 L16-17: Specify what types of aerosol this enhancement applies to

P1 L25: Reference desired for this information.

P1 L26-28: HG reference

P1 L27: "The HG phase function $(P_{HG}(\theta))$ is..."

P2 L34: "$P(\theta)$ is the *normalized* scattering phase function"

P2 L37: More references would be useful here.

P2 L39: "few studies" – which ones?

P2 L49: What is meant by "no available method to measure g directly" – surely a measurement of $P(\theta)$ is a fairly direct measurement technique? (i.e. Bian, Dolgos instruments)

P2 L55: Define backward hemisphere angles

P2 L56: It would be appropriate perhaps to also cite Horvath et al, 2016 (J. Aerosol Sci.)

P3 L57-63: How were these values determined – observations (if so, which instruments used?) or models?

P3 L64-66: Can you discuss difficulties measuring g under ambient RH vs dry conditions?

P3 L64-66: It would be appropriate to cite Andrews et al. again here

P3 L68: Particles do not absorb water –they can take up water, or water can condense on them

P3 L73-76: Define $M_{water}$

P3 L79-80: Should specify that Mie theory only applies to spherical particles – g can also vary by morphology

P3 L84: Can you use subscript for $f_{RH}$ to reduce confusion (instead of fRH)?

P4 L87: Define back-scattering coefficient

P4 L106-110: Please provide seasonal information about these field campaigns, as well as whether they covered the full diurnal cycle.

P4 L110-111: For people who are unfamiliar with Beijing, please describe the location in terms of either distance from the centre or relative population density – some more general metric

P4 L110-P5 L116: Can you also describe the general wind patterns (i.e. are the sites downwind or upwind of the urban centres?)

P5 L118-119: Type of impactor? Is 10 μm the 50% cut-point?

P5 L119: How is RH measured?

P5 L121-122: Are particles assumed to be spherical for SMPS and APS retrievals of size?

P5 L119: How low was RH? How sensitive are results to variations in RH at these low levels (presumably it is not sensitive due to low gf in this region but would be useful to say this explicitly)?

P5 L119-132: Please provide error/uncertainty margins on key instrumentation (i.e. RH ±1%?)

P5 L129: Provide laser wavelength and polarization characteristics (circular or linear)

P5 L126: Please be consistent about f(RH) vs fRH

P5 L127: This is a very long measurement period – can you ignore changes in aerosol population over this timescale?

P5 L128: What is time resolution of CCD-LADS?

P5 L136: Please clarify BC mixing state and BC fraction determination earlier.

P5 L127: Need more details about "self-constructed humidified nephelometer system" – what range/steps of RH? Is the population sufficiently consistent? Is 45 min for range of RH or single RH set point? How was RH altered? How was RH monitored? Is there another reference with more details about this instrument?

P6 L148: Why use volume mixing ratio vs, for example, Maxwell-Burnett or mass?

P6 L150: If using a core-shell model with BC in the centre, do you assume water only mixes with shell, or with both core and shell?

P6 L147: How is the corresponding ambient aerosol PNSD at given RH computed?

P6 L157-158: Can you provide references going into more detail about random forest model algorithms?

P6 L158-162: Why is the random forest model appropriate for this specific example?

P6 L164: How was the number of trees determined? What is the sensitivity of the results to $n_{pre}$ and $n_{tree}$?

P6 L165: scattering coefficients for dry aerosol? Or humidified? If humidified, isn't there a relationship between $\kappa$ and the scattering coefficients?

P6 L163-168: Can you provide more justification for your chosen parameterization? Does the connection between measured scatter and backscatter coefficients affect the suitability of the algorithm, or the accuracy of the results?

P L170-171: Why did you use separate data sets for training vs testing? Why not use a subset of each, or subset of one and test on the other subset?

P6 L173: Specify humidified vs dry nephelometer

P7 L189-191: What is ceiling (maximum altitude) of your SBDART model?

P7 L191-192: Assume HG phase functions or Mie phase functions in SBDART?

P7 L194-195: Mean results from a specific span of time? Different times of day? Please be more specific

P7 L199: Please provide reference for SBDART defaults values

P7 L192-199: I am a bit confused – were radiosonde data from the *summer* combined with albedo data from the *winter*? Can you explain?

P8 L232: Does setting RH and $\kappa$ values give different results from removing them as input variables and setting $n_{pre}=6$?

P9 L238-239: Sensitivity of calculated g values to input RH and $\kappa$?

P9 L247: Should mention that other empirical relationships have been given between g and b (i.e. Sheridan and Ogren, 1999 (JGR); Moosmüller and Ogren, 2017 (Atmos.); Marshall et al, 1995 (Appl. Opt.)) Perhaps one of the others is more accurate in these cases?

P9 L256-261: Why random/arbitrary deviations in input values? Can you use the specified/measured uncertainties in the measurements themselves?

P10 L269: Based on your own comments earlier in the paragraph isn't the model simply more sensitive to completely independent input variables, which happen to be RH and κ in this example? What happens if you only input $\sigma_{sca}$, or specify that $\sigma_{sca}$ and $\beta_{sca}$ must vary in the same way?

P10 L274-276: Can you clarify what you mean by "actual calculated aerosol phase function" – does this imply those measured directly by CCD-LADS, or those calculated using Mie theory? If the latter, can you also compare DARF using directly measured phase functions using CCD-LADS?

P10 L289: Suggest citing McComiskey et al, 2008 (JGR)

P10 L289-290: It is unclear how this section ties into the rest of the paper, or what novelty comes from the discussion. As you point out, previous studies have also undertaken to study how modelled DARF varies with input parameters like asymmetry parameter. Can you add anything unique to the literature regarding g and DARF?

P10 L290-298: Would it be perhaps more appropriate to vary g according to a typical RH profile? You show that g tends to be higher when RH >90%, conditions which are only likely to occur in specific layers of the atmosphere depending on the local meteorology.

P10 L291: Where does 2.3% come from?

P12 L339: Regarding "actual phase function", please clarify (see general comments)

Figure 1: What about Hauirou? Can you also provide graphs of average PNSDs for each site?

Figure 3: Clarify which data used for these calculations – which site.

Figure 4: Please explain in text a possible justification for the higher dependence on $\sigma_{sca,635}$ compared to other wavelengths. Also, why is g very dependent on 5% variation in κ, but not significantly affected by greater deviations?

Figure 5: Clarify "calculated phase function" (see general comments)

Figure 7: Please clarify figure in caption by changing order: $g_{CCD,Cal}$ as a function of $g_{Machine,cal}$

**Technical comments**

P6 L157: Please correct "tress" to "trees"

---

## Referee Comment (RC2) · Anonymous Referee #1 · 6 Apr 2018

The paper 'Method to calculate the aerosol asymmetry factor based on measurements from the humidified nephelometer system' offers a new method for determining the ambient aerosol asymmetry factor. I believe that aerosol asymmetry factor is clearly of real importance. The proposed method has advantages over the traditional methods as it can measure the aerosol asymmetry factor in real time. I am glad to see the results of the effects of aerosol hygroscopic growth on the variation in aerosol asymmetry factor, which is rarely discussed in previous studies. Overall, the paper is clearly written and contains originality. I recommend that the paper be accepted for publication in ACP after some minor work to be done for its improvements. I have the following suggestions to further improve this work: (1) The authors should highlight the novel

and original aspects of the work. More discussions should be added in the text, mostly the introduction section. (2) To my knowledge, the aerosol asymmetry factor is highly related to the aerosol particle number size distribution, the aerosol mixing states, the ambient relative humidity (RH) and the aerosol complex refractive index. The first three parameters are discussed in this work, I suggest that the authors add some work on the sensitivities of the aerosol asymmetry factor on complex refractive index. The uncertainties due to complex refractive index should be well discussed in this paper. (3) The method of training the machine learning model should be reconstructed. There are large uncertainties for measurements of particle number size distribution. I suggest that it would be better if the authors use all of the training data from the calculations of the Mie scattering model with measured particle number size distributions. In this way, the aerosol scattering coefficient, aerosol backscattering coefficient and the aerosol asymmetry factor under different RH can be calculated using the measurements of particle number size distributions, the mass concentration of the black carbon and aerosol hygroscopic growth factor, $\kappa$. This can avoid the uncertainties in measurements of the aerosol particle number size distributions. (4) In section 3.3, parameterization of the aerosol vertical profiles of the aerosol optical properties should be discussed in detail. (5) Section 4.4 gives the validation of the random forest machine model, it should be placed after section 4.2. (6) I suggest that figure 6 re-plotted and be presented in a clearer way. (7) Line 114 : What is the meaning of UCAS? Please describe it. (8) Line 312: 'to' should be changed to 'in'
* * *

---

## Author Comment (AC1) · 11 May 2018

All of the documents are zipped in the supplementary file.

Please also note the supplement to this comment:
https://www.atmos-chem-phys-discuss.net/acp-2017-1148/acp-2017-1148-AC1-supplement.zip

---

## Author Comment (AC2) · 11 May 2018

All of the documents are zipped in the supplementary file.

Please also note the supplement to this comment:
https://www.atmos-chem-phys-discuss.net/acp-2017-1148/acp-2017-1148-AC2-supplement.zip